# Lycoperoside H, a Tomato Seed Saponin, Improves Epidermal Dehydration by Increasing Ceramide in the Stratum Corneum and Steroidal Anti-Inflammatory Effect

**DOI:** 10.3390/molecules26195860

**Published:** 2021-09-27

**Authors:** Shogo Takeda, Kenchi Miyasaka, Sarita Shrestha, Yoshiaki Manse, Toshio Morikawa, Hiroshi Shimoda

**Affiliations:** 1Research and Development Division, Oryza Oil and Fat Chemical Co., Ltd., 1 Numata, Kitagata-cho, Ichinomiya 493-8001, Aichi, Japan; kgohedasato2@gmail.com (S.T.); kgohedasato3@gmail.com (K.M.); 2Pharmaceutical Research and Technology Institute, Kindai University, 3-4-1 Kowakae, Higashi-osaka 577-8502, Osaka, Japan; japan.sarita@gmail.com (S.S.); manse@phar.kindai.ac.jp (Y.M.); morikawa@kindai.ac.jp (T.M.)

**Keywords:** tomato seed, lycoperoside, steroidal saponin, ceramide, transepidermal water loss, anti-inflammation

## Abstract

Tomatoes are widely consumed, however, studies on tomato seeds are limited. In this study, we isolated 11 compounds including saponins and flavonol glycosides from tomato seeds and evaluated their effects on epidermal hydration. Among the isolated compounds, tomato seed saponins (10 µM) significantly increased the mRNA expression of proteins related to epidermal hydration, including filaggrin, involucrin, and enzymes for ceramide synthesis, by 1.32- to 1.91-fold compared with the control in HaCaT cells. Tomato seed saponins (10 µM) also decreased transepidermal water loss by 7 to 13 g/m^2^·h in the reconstructed human epidermal keratinization (RHEK) models. Quantitative analysis of the ceramide content in the stratum corneum (SC) revealed that lycoperoside H (1–10 µM) is a promising candidate to stimulate ceramide synthesis via the upregulation of ceramide synthase-3, glucosylceramide synthase, and β-glucocerebrosidase, which led to an increase in the total SC ceramides (approximately 1.5-fold) in concert with ceramide (NP) (approximately 2-fold) in the RHEK models. Evaluation of the anti-inflammatory and anti-allergic effects of lycoperoside H demonstrated that lycoperoside H is suggested to act as a partial agonist of the glucocorticoid receptor and exhibits anti-inflammatory effects (10 mg/kg in animal test). These findings indicate that lycoperoside H can improve epidermal dehydration and suppress inflammation by increasing SC ceramide and steroidal anti-inflammatory activity.

## 1. Introduction

Tomato (*Solanum lycopersicum*) is a popular food consumed around the world. It contains various nutritional phytochemicals, including vitamins, carotenoids, saponins, and flavonoids [1]. Among these, previous works regarding tomato saponins are limited. The spirosolane types of steroidal saponins have been isolated as lycoperosides [2,3] and escleosides [4,5,6,7] from tomato. Regarding the biological activities of these steroidal saponins, Fujiwara et al. have reported that the oral administration of esculeoside A reduced serum cholesterol and low-density lipoprotein-cholesterol in apolipoprotein E-deficient mice [8]. On the other hand, Zhou et al. have reported that orally administered esculeoside B isolated from tomato juice improved 2,4-dinitrochlorobenzene-induced type IV allergic dermatitis in mice [9]. However, research specific to the seeds is limited.

We had an opportunity to obtain dried tomato seeds, and hence we performed a chemical study on low molecular components in the tomato seeds and found that they contain several types of saponins and flavonol glycosides (Figure 1). Among them, we have recently demonstrated that oral administration of lycoperoside H, a steroidal saponin isolated from tomato seeds, ameliolated atopic dermatitis (AD)-like skin inflammation and transepidermal water loss (TEWL), in concert with a decrease in accumulation of mast cells, eosinophils in dermis, the secretion of serum total IgE, and the Th2/Th1 cytokine ratio in IL-33 transgenic mice [10]. However, the mechamism of lycoperoside H in this study was not evaluated in detail and the anti-inflammtory and/or epidermal hydrating effect of isolated compounds other than lycoperoside H remain unknown.

Generally, saponins exhibit a variety of biological activities. Among them, the anti-inflammatory effects [11] and wound healing effects [12] of saponins on human skin have been recognized as beneficial effects for decades. For example, Korean ginseng is well-known for containing specific saponins, ginsenosides. Lee et al. have reported that the oral administration of Korean ginseng extract containing ginsenosides improved TEWL in AD mice, meaning an increase in anti-inflammatory effects and skin hydration [13]. Especially, ginsenoside Rg1, a principal ginsenoside, ameliorated the increase in TEWL in hairless mouse skin damaged by UV rays [14]. In addition to the anti-inflammatory effects, ginsenoside Rc enhances filaggrin expression in HaCaT cells, a keratinocyte cell line, which contributes to the retention of epidermal moisture [15]. On the other hand, gracillin, a spirostan-type saponin, isolated from *Dioscorea quinqueloba,* improved TEWL in an AD mouse model [16], and centella saponins in the leaves and stems of *Centella* (*C*) *asiatica* are famous for their wound healing effect by promoting collagen production [17]. Centella saponins also reduce epidermal dehydration. Madecassoside, a major pentacyclic triterpene saponin in *C. asiatica*, reduces TEWL and enhances several moisturizing molecules, including aquaporin-3, loricrin, and involucrin [17]. Moreover, topical application of a cosmetic formulation containing *C. asiatica* extract has been used for skin moisturizing and anti-inflammation [18].

In consideration of the above references, we built the hypothesis that tomato seed extract (TSE) and tomato seed saponins affect epidermal hydration by improving TEWL or increasing moisturizing factors. Since several factors, including ceramide, a dominant lipid in the stratum corneum (SC), filaggrin, and involucrin, contribute to epidermal hydration, we conducted a study on the epidermal hydrating effect of tomato seed saponins including lycoperoside H and flavonoids through epidermal molecule expressions contributing toward hydration and TEWL suppression.

## 2. Results

### 2.1. Effect of TSE and Isolated Compounds (1–11) on the mRNA Expression of Proteins Related to Epidermal Hydration in HaCaT Cells 

We evaluated the effect of isolated compounds from tomato seeds on the mRNA expression of filaggrin, involucrin, serine palmitoyltransferase-2 (SPT2), ceramide synthase-3 (CerS3), and glucosylceramide synthase (GCS), which play crucial roles in epidermal hydration. As shown in Table 1, TSE significantly upregulated the mRNA expression of involucrin, SPT2, CerS3, and GCS, and tended to increase that of filaggrin. Similar to TSE, lycoperoside C (**2**) significantly increased the mRNA expression of all the proteins examined. Lycoperoside H (**3**) significantly enhanced the mRNA expression of proteins, except filaggrin and SPT2. On the other hand, lycoperoside A (**1**) did not affect any mRNA expression of proteins. As for allopregnenolone 3-*O*-β-solatrioside (**4**), only significant upregulation of involucrin was observed. 

Tomatoside A (**6**) significantly increased CerS3 and GCS expression, and tigogenin 3-*O*-β-solatrioside (**7**) enhanced SPT2 and GCS expression. In contrast to the effects of tigogenin 3-*O*-β-solatrioside (**7**), 22α-methoxytomatoside A (**8**) significantly increased the mRNA expression of filaggrin and CerS3. In terms of the flavonoids, only naringenin (**5**) and astragalin (**9**) exhibited significant upregulation of SPT2, whereas rutin (**10**) significantly downregulated the mRNA expression of CerS3 and GCS, and quercetin 3-*O*-β-cellobioside (**11**) significantly decreased the expression of filaggrin, SPT2, CerS3, and GCS and tended to diminish involucrin expression.

### 2.2. Effects of TSE and Saponins (**1**–**4**,**6**–**8**) on TEWL in RHEK Models 

As a result of the screening for active compounds for skin hydration using HaCaT cells, tomato seed saponins (**1**–**4**,**6**–**8**) were selected as the candidates which upregulate the mRNA expression of factors related to epidermal hydration. As a next step, we evaluated the effect of TSE and tomato seed saponins (**1**–**4**,**6**–**8**) on TEWL in RHEK models and whether they have hydrating activity. TSE (10 µg/mL) significantly decreased TEWL on days 5 and 7 (Figure 2A). As for the tomato seed saponins, 10 µM of **1**, **3**, and **7** significantly suppressed TEWL on days 5 and 7 (Figure 2A,B). Compound **8** significantly decreased TEWL on day 5 and **2** and **6** reduced TEWL on day 7 (Figure 2B). On the other hand, **4** had no effect on TEWL.

### 2.3. Effects of Tomato Seed Saponins (**1**–**4**,**6**–**8**) on the Ceramide Contents in SC of RHEK Models

Figure 3A shows the effects of tomato seed saponins (**1**–**4**,**6**–**8**) on SC ceramide contents in RHEK models. Among tomato seed saponins, only lycoperoside H (**3**) significantly increased the total ceramide contents, and tigogenin 3-*O*-β-solatrioside (**7**) slightly increased the total ceramide contents but the difference was not significant. In terms of the ceramide species, ceramide (NP) was significantly increased by lycoperoside H (**3**) and tended to increase with treatment of tigogenin 3-*O*-β-solatrioside (**7**) and 22α-methoxytomatoside A (**8**). Although similar trends were observed in the contents of ceramide (NS, NDS), (EOH), (AS), and (AP), significant differences were not confirmed. Other tomato seed saponins did not affect the SC ceramide contents. 

As described above, only lycoperoside H (**3**) significantly increased the SC ceramide contents at 10 µM among the tomato seed saponins (**1**–**4**,**6**–**8**). Thus, we next evaluated the concentration dependency of lycoperoside H (**3**). The effect of 1 and 10 µM of lycoperoside H (**3**) on the SC ceramide contents in RHEK models is shown in Figure 3B. Lycoperoside H (**3**) increased the total ceramide contents dose-dependently, and a significant difference was observed at 10 µM. In addition, lycoperoside H (**3**) significantly increased the ceramide (NP) contents at 1 and 10 µM with dose dependency. As for the ceramide (NS, NDS), (EOS), (EOH), (AS), and (AP), lycoperoside H (**3**) tended to increase their contents.

### 2.4. Effects of Lycoperoside H (**3**) on the mRNA Expression of Enzymes Related to Ceramide Synthesis in RHEK Models

Lycoperoside H (**3**) significantly increased the SC ceramide contents in RHEK models, as described above. Therefore, we evaluated the effects of lycoperoside H (**3**) on the expression of enzymes involved in SC ceramide synthesis. Figure 4 shows the effects of lycoperoside H (**3**) on the mRNA expression of enzymes related to SC ceramide synthesis, SPT2, CerS3, GCS, β-glucocerebrosidase (GBA), sphingomyelin synthase-2 (SMS2), and acid sphingomyelinase (ASM) in RHEK models. Lycoperoside H (**3**) significantly upregulated the mRNA expression of CerS3, GCS, and GBA at 1, 3, and 10 µM, 3 and 10 µM, and 10 µM, respectively. On the other hand, the mRNA expression of SPT2 and SMS2 were significantly downregulated by the treatment of lycoperoside H (**3**) at 1, 3, and 10 µM. No significant change was observed in the mRNA expression of ASM.

### 2.5. Anti-Inflammatory and Anti-Allergic Effects of Lycoperoside H (**3**)

The anti-inflammatory and anti-allergic effects of lycoperoside H (**3**) were evaluated to clarify the mechanism that contributes to the suppression of skin barrier deterioration, other than ceramide production enhancement. TSE and lycoperoside H (**3**) did not affect the compound 48/80-induced and histamine-induced scratching behavior in mice (Figure 5A). On the other hand, significant suppression of the IgE-mediated passive cutaneous anaphylaxis (PCA) reaction in mice was observed with a high dose of TSE administration, but not with lycoperoside H (**3**) (Figure 5B). Figure 5C shows the effects of TSE, lycoperoside H (**3**), and tigogenin 3-*O*-β-solatrioside (**7**) on the acetic acid-induced writhing (upper) and vascular permeability (lower) in mice. TSE and lycoperoside H (**3**) significantly suppressed writhing and vascular permeability, while tigogenin 3-*O*-β-solatrioside (**7**) did not show any effects. As a result of the glucocorticoid receptor competitive assay, lycoperoside H (**3**) exhibited partial glucocorticoid receptor binding ability (Figure 5D). Figure 5E shows the effects of TSE and lycoperoside H (**3**) on histamine-induced guinea pig tracheal muscle contraction. TSE and lycoperoside H (**3**) did not show an antihistaminic effect in tracheal muscle contraction.

## 3. Discussion

In previous studies on tomato saponins, steroidal saponins, such as lycoperosides (**1**–**3**), had been isolated from whole fruits [2,19,20,21]. Furthermore, flavonoids including naringenin (**5**), astragalin (**9**), and rutin (**10**) had been reported to exist in the fruits [22]. On the other hand, as for the constituent study of tomato seeds, these compounds had not been isolated, while hydrophobic compounds, such as β-sitosterol, stigmasterol, and γ-tocopherol, have been found [23]. In the present study, we isolated 7 saponins and 4 flavonoids from tomato seeds. Thus, our findings may be the first report that tomato seeds contain steroidal saponins and flavonoids as secondary metabolites.

Aiming to find the candidate compounds for skin hydration, the effects of 11 isolated compounds on the mRNA expression of proteins related to epidermal hydration, filaggrin, involucrin, SPT2, CerS3, and GCS were evaluated. Filaggrin provides natural moisturizing amino acids by its self-decomposition in SC [24] and acts as a filament-aggregating protein [25]. Involucrin is one of the essential proteins for the formation of the cornified envelope which contributes to the barrier function of the skin [26,27]. SPT2, CerS3, and GCS are involved in the synthesis of SC ceramides, which exist as dominant lipids in the SC of human skin and play important roles in the moisturization and barrier function [28]. SPT and CerS are involved in the de novo synthesis of ceramides [29,30] and GCS acts as a glucosyltransferase enzyme to synthesize glucosylceramides, which are precursors of SC ceramides [31]. In the present study, tomato seed saponins (**1**–**4**,**6**–**8**), especially lycoperoside C (**2**), significantly upregulated the mRNA expression of these enzymes in HaCaT cells. On the other hand, tomato seed flavonoids (**5**,**9**–**11**) exhibited a trend to downregulate the expression of enzymes for ceramide synthesis. These results suggest that tomato seed saponins appeared to be leading candidates that contribute to epidermal hydration. Regarding previous reports of saponins on skin health, Oh et al. reported that ginsenoside Rc isolated from *Panax ginseng* protects the epidermis from UVB-induced photooxidative damage with the upregulation of filaggrin expression in HaCaT cells [15]. Likewise, madecassoside, an ursane-type saponin from *C. asiatica*, has been reported to increase the mRNA and protein expressions of involucrin in HaCaT cells [17]. These findings may support our present findings of the hydrating effects of tomato seed saponins observed in HaCaT cells. In terms of the flavonols (**9**–**11**), we have demonstrated that tiliroside, a flavonol glycoside isolated from strawberry seeds, upregulated the expression of GCS and increased SC ceramide [32]. Regarding this finding, which is contrary to the present result, the presence of coumaroyl moiety in the structure of tiliroside is suggested to be involved in these effects. Li et al. demonstrated that tiliroside exhibits much greater antioxidant and cytoprotective activities compared to astragalin (**9**), which has the same structure, other than the coumaroyl moiety [33]. Thus, it is considered that the difference between tomato seed flavonoids and tiliroside in the effect on GCS expression is also due to the presence or absence of the coumaroyl moiety. 

TEWL is the most frequently used parameter for the evaluation of the epidermal barrier function [34]. Significant elevation of TEWL is commonly observed in dry skin diseases such as AD [35], xerosis, and psoriasis [36]. To measure TEWL in vitro and in vivo, several TEWL devices have been developed. Recently, Tewitro TW24, a device which can directly and simultaneously measure the TEWL of RHEK models on a 24-well plate, has been developed [37,38]. However, there are no reports that evaluated the effect of natural compounds on TEWL using this device. Thus, we used this device to assess the effects of tomato seed saponins on changes in TEWL in RHEK models in this study. As a result of the measurements, all tomato seed saponins (**1**–**3**,**6**–**8**) except **4** significantly decreased TEWL during the 7 days of treatment. Considering the difference in the effect between **4** and the other saponins, it is suggested that the difference in aglycon structure might be involved. Allopregnenolone 3-*O*-β-solatrioside (**4**) has a pregnane steroid structure which consists of only steroidal A- to D-rings. On the other hand, saponins (**1**–**3**,**6**–**8**) which affected the TEWL have a furostanol, a spirosolanol, or a spirostanol steroid structure which consist of not only steroidal A- to D-rings but also an E-ring or E- to F-rings [39]. Similarly, it has been reported that topical application of gracillin, a spirostanol-type steroidal saponin isolated from *Dioscorea quinqueloba*, prevented TEWL elevation on 2,4-dinitrochlorobenzen-induced AD-like skin in mice [16]. Gracillin also has steroidal A- to F-rings. Therefore, it is suggested that the TEWL-lowering effect of saponins requires at least an E-ring in the structure.

Ceramides are a family of sphingolipids consisting of a sphingoid base and a fatty acid. Ceramides dominantly exist in the SC and play pivotal roles as a water reservoir and a barrier [28]. In the profiling study of ceramide species, 12 major classes of ceramides have been found in the human SC [40]. SC ceramide significantly decreased in patients with AD [41] or xerosis [42], and decreasing ceramide correlates to epidermal water evaporation [43]. Among the tomato seed saponins, only lycoperoside H (**3**) significantly increased the total SC ceramide contents in the RHEK models in concert with a significant increase of ceramide (NP). Lycoperoside H (**3**) is the most abundant saponin in TSE, and its concentration-dependent effect (1 and 10 µM) was observed in ceramide production. Our previous studies demonstrated that tiliroside isolated from strawberry seeds increases the ceramide (NS, NDS) contents [32], and β-sitosterol 3-*O*-glucoside isolated from rice bran increased ceramide (EOS) [44]. On the other hand, the present study demonstrated that lycoperoside H (**3**) increased ceramide (NP), which was not observed in tiliroside and β-sitosterol 3-*O*-glucoside. Therefore, the mechanism of increasing SC ceramide may be different to tiliroside and β-sitosterol 3-*O*-glucoside. 

Since a positive effect of lycoperoside H (**3**) on the SC ceramide contents was observed, we evaluated the effects of lycoperoside H (**3**) on the mRNA expression of enzymes related to SC ceramide synthesis in the RHEK models. In addition to SPT2, CerS3, and GCS, as mentioned above, GBA, SMS2, and ASM are involved in SC ceramide synthesis. GBA hydrolyzes glucosylceramide, which is synthesized by GCS to SC ceramide [45]. SMS catalyzes the synthesis of sphingomyelin [46] and ASM synthesizes the SC ceramide from sphingomyelin [47]. Lycoperoside H (**3**) significantly upregulated the mRNA expression of CerS3, GCS, and GBA. On the other hand, significant downregulation of the mRNA expression of SPT2 and SMS2 was also observed. The ceramide synthesis pathway in the epidermis was regulated by two distinct streams, which involve glucosylceramide or sphingomyelin as intermediate products, and a major part of SC ceramide species are synthesized from glucosylceramides [31]. Thus, it is considered that the upregulation of GCS and GBA by lycoperoisde H (**3**) mainly led to an increase in SC ceramide. However, the effect of downregulation of SPT2, which is the most upstream enzyme of the pathway, remains unclear in this study. Taken together, these results suggest that lycoperoside H (**3**) is a promising compound to increase the level of SC ceramide by the upregulation of CerS3, GCS, and GBA expression, in turn leading to TEWL improvement. 

Although lycoperoside H (**3**) significantly increased the SC ceramide contents, we thought that only the decrease in TEWL was not sufficient to ameliorate severe AD inflammation, as previously reported [9,11]. Therapy with steroidal medicines improves TEWL due to anti-inflammatory and anti-allergic effects [48,49]. Therefore, we evaluated whether lycoperoside H (**3**) has anti-inflammation and/or anti-allergic effects. As a result, lycoperoside H (**3**) did not exhibit suppressive effects on compound 48/80-induced pruritus and IgE-mediated PCA reaction in mice. From these results, lycoperoside H (**3**) is not considered to be effective against type-Ⅰ allergy involving histamine as a mediator. Actually, lycoperoside H (**3**) did not affect histamine-induced scratching behavior in mice and histamine-induced guinea pig tracheal muscle contraction. These results suggest that lycoperoside H (**3**) is not an antihistaminic agent. On the other hand, lycoperoside H (**3**) significantly suppressed acetic acid-induced writhing behavior and vascular permeability in mice, while tigogenin 3-*O*-β-solatrioside (**7**) did not exhibit any effect. Similarly, escins, the oleanane-type saponins isolated from the seeds of *Aesculus hippocastanum* L., have been reported to show a suppressive effect on acetic acid-induced vascular permeability in mice [50]. Shehu et al. also reported that the saponin-rich fraction of *Laggera aurita* suppressed the acetic acid-induced writhing [51]. Thus, the mechanism of lycoperoside H (**3**) in this experiment is considered similar to that of these saponins. Finally, as a result of the glucocorticoid receptor competitive assay, the partial glucocorticoid receptor binding ability of lycoperoside H (**3**) was observed. A previous report demonstrated that ginsenoside Rg1, a steroidal saponin derived from *Panax ginseng*, was shown to fully bind to the human glucocorticoid receptor in the glucocorticoid receptor competitive assay, and docking simulation revealed that the ginsenoside molecule marginally fits into the hydrophobic cavity around the ligand-binding domain of the glucocorticoid receptor, in spite of the sugar moiety causing steric bumps [52]. Ginsenoside Rg1 has steroidal A- to D-rings with 2 sugar moieties which bind to A- and B-rings, respectively. Similarly, lycoperoside H also has steroidal A- to D-rings with 4 sugars which bind to the A-ring. Thus, it is reasonable to understand that both compounds could exhibit steroidal activity through the glucocorticoid receptor, and lycoperoside H has less and partial ability to bind to the glucocorticoid receptor because of disrupted binding by 4 sugars on the A-ring. In the other study, the glucocorticoid receptor binding assay using FTO2B rat hepatoma cells demonstrated that ginsenoside Rg1 binds to the glucocorticoid receptor with 1/10 to 1/100 ability of Dx [53]. Therefore, these findings suggest that lycoperoside H (**3**) is a partial agonist of the glucocorticoid receptor and acts as a natural steroidal agent to exhibit anti-inflammatory effects. Taken together, it is suggested that lycoperoside H (**3**) improved TEWL by acting as a partial glucocorticoid receptor agonist with suppression of the acute inflammatory effect. This activity is also considered to be the mechanism of our previous results, which demonstrated that oral administration of lycoperoside H suppressed AD-like skin inflammation in IL-33 transgenic mice [10].

In conclusion, we identified 11 compounds from tomato seeds, and among them, lycoperoside H (**3**) has the potential to improve TEWL by the SC ceramide increasing effect, in concert with significantly increased ceramide (NP) contents and by anti-inflammatory effects via acting as a partial glucocorticoid receptor agonist. Therefore, lycoperoside H (**3**) is a promising skin hydrating and anti-inflammatory compound that can be expected to have excellent effects on epidermal hydration.

## 4. Materials and Methods

### 4.1. Preparation of Tomato Seed Extract and Its Compounds

To isolate lycoperoside A (**1**), lycoperoside C (**2**), lycoperoside H (**3**), allopregnanolone 3-*O*-β-solatrioside (**4**), and naringenin (**5**), tomato seeds (2 kg) obtained from JiuQuan Jiuzhou Seed Co., Ltd. (Gansu, China) were defatted with hexane. The defatted seeds were ground and MeOH (10 L) was used for extraction at 70 °C for 3 h, and then the solvent was evaporated to yield the tomato seed extract (TSE, 44 g, yield 2.2%). The TSE was suspended in water (500 mL) and extraction was performed with EtOAc (500 mL, twice) and *n*-BuOH (500 mL, twice). These solvents were evaporated to obtain the EtOAc layer (0.63 g, yield 0.03%), *n*-BuOH layer (2.41 g, yield 0.12%), and H_2_O layer (5.05 g, yield 0.26%). The *n*-BuOH layer underwent silica-gel column chromatography [CHCl_3_:MeOH (9:1→7:3)→CHCl_3_:MeOH:water (6:4:1)→MeOH] to obtain fraction (Fr.) 1 (183.4 mg), Fr.2 (623.5 mg), Fr.3 (1.12 g), and Fr.4 (246.4 mg). Fr.3 was purified by reversed-phase HPLC (Inertsil Ph-3, 20 φ × 250 mm; GL Science Inc., Tokyo, Japan) with 70% MeOH to obtain lycoperoside A (**1**, 9.4 mg, yield 0.0005%), lycoperoside C (**2**, 1.9 mg, yield 0.00001%), lycoperoside H (**3**, 12.2 mg, yield 0.0006%), and allopregnenolone 3-*O*-β-solatrioside (**4**, 11.2 mg, yield 0.0006%). The EtOAc layer was purified by reversed-phase HPLC (Inertsil ODS-SP, 20 φ × 250 mm; GL Science Inc. Tokyo, Japan) with 80% MeOH to obtain naringenin (**5**, 3.3 mg, yield 0.0002%). 

To isolate tomatoside A (**6**), tigogenin 3-*O*-β-solatrioside (**7**), 22α-methoxytomatoside A (**8**), astragalin (**9**), rutin (**10**), and quercetin 3-*O*-β-cellobioside (**11**), the H_2_O layer (60.0 g) after EtOAc extraction underwent Diaion^®^ HP-20 column chromatography (water→MeOH) to obtain the MeOH-eluted Fr. (22.94 g, yield 2.15%) and the water-eluted Fr. (33.18 g, yield 3.11%). The MeOH-eluted Fr. was fractionated by ODS column chromatography (40% MeOH→60% MeOH→90% MeOH→MeOH) to obtain Fr.1 (1.47 g), Fr.2 (0.82 g), Fr.3 (1.59 g), Fr.4 (0.77 g), Fr.5 (10.37 g), Fr.6 (5.49 g), Fr.7 (2.06 g), and Fr.8 (0.84 g). Fr. 8 was purified by reversed-phase HPLC (Cosmosil 5C_18_-MS-Ⅱ, 20 φ × 250 mm; Nacalai Tesque Inc., Kyoto, Japan) with 90% MeOH–1% acetic acid to obtain tomatoside A (**6**, 12.0 mg, yield 0.0034%) and tigogenin 3-*O*-β-solatrioside (**7**, 19.5 mg, yield 0.0055%). Fr. 5 was purified by reversed-phase HPLC (Cosmosil 5C_18_-MS-Ⅱ, 20 φ × 250 mm) with 50% MeOH–1% acetic acid to obtain 22α-methoxytomatoside A (**7**, 289.9 mg, yield 0.939%). Fr. 3 was purified by reversed-phase HPLC (Cosmosil 5C_18_-MS-Ⅱ, 20 φ × 250 mm) with 50% MeOH–1% acetic acid to obtain astragalin (**9**, 31.0 mg, 0.0247%), rutin (**10**, 10.0 mg, 0.0080%), and quercetin 3-*O*-β-cellobioside (**11**, 14.6 mg, 0.0112%). The chemical structures of the isolated compounds were identified by a comparison of the ^1^H- and ^13^C- NMR spectra with References [2,19,20,54,55,56,57,58,59,60]. The chemical structures of the isolated compounds (**1**–**11**) are shown in Figure 1.

### 4.2. Reagents

Dulbecco’s modified Eagle medium (DMEM), 0.25 w/v% trypsin 1 mmol/L/EDTA 4Na solution with phenol red (trypsin/EDTA aqueous solution), phosphate-buffered saline (PBS), skim milk, DP, DX, histamine dihydrochloride, gum Arabic, Evans blue, and pontamine sky blue were obtained from FUJIFILM Wako Pure Chemical Co. Ltd. (Osaka, Japan). The RNeasy^®^ Mini Kit was purchased from QIAGEN (Hilden, Germany). PrimeScript™ Reverse Transcriptase and TB Green^®^ Premix Dimer Eraser™ were purchased from Takara Bio Inc. (Kusatsu, Japan). The dNTP mixture, random primer, and PolarScreen™ Glucocorticoid Receptor Competitor Assay Kit, Red, were purchased from Invitrogen (Waltham, MA, USA). HPTLC plate and DNP-BSA were purchased from Merck Millipore (Darmstadt, Germany). Fetal bovine serum (FBS) was purchased from Biosera (Boussens, France). The ceramide standards of ceramide (NS, NDS) and (AS) were purchased from Matreya LLC. (Philadelphia, PA, USA). Anti-DNP IgE was obtained from Seikagaku Industry (Tokyo, Japan).

### 4.3. Cells and Animals

Immortalized human keratinocytes, HaCaT cells, were kindly supplied by Kindai University (Osaka, Japan). RHEK models (LabCyte EPI-MODEL) obtained from Japan Tissue Engineering Co., Ltd. (Gamagori, Japan) were used for the measurement of TEWL, quantification of ceramide, and enzymes related to ceramide synthesis expression. LabCyte EPI-MODEL was used for the measurement of TEWL, and LabCyte EPI-MODEL 6D, which was cultured under pre-keratinization conditions before the formation of the SC layer, was used for the other experiments.

Different animal sources were selected according to the experimental method. Male ddY mice aged 5 and 8 weeks old and male Hartley guinea pigs aged 5 weeks old were purchased from Japan SLC Inc. (Hamamatsu, Japan). The mice and guinea pigs were acclimated for 7 days under 22 ± 2 °C and 50% ± 5% RH before the experiments and were fed a standard CE-2 non-purified diet (Clea Japan Inc., Shizuoka, Japan) and LRC4 (Oriental Yeast Co. Ltd., Tokyo, Japan), respectively. The animal experiments were performed in accordance with the Guidelines for Animal Experimentation (Japan Association for Laboratory Animal Science, 1987). All the animal experiments were approved by the ethics committee of Oryza Oil and Fat Chemical Co., Ltd. (Aichi, Japan).

### 4.4. Culture of HaCaT Cells for the Screening of Candidate Compounds with Hydration Activity

HaCaT cells (1.0 × 10^5^ cells/well) were seeded onto a 24-well culture plate and maintained in DMEM with 10% FBS at 37 °C, 5% CO_2_ atmosphere. After incubation for 1 night, the medium was changed to DMEM without FBS and incubated for 24 h, and the cells were treated with TSE (10 μg/mL) or isolated compounds (10 μM) which were dissolved in dimethyl sulfoxide (DMSO) for 6 h. The final concentration of DMSO was adjusted to 0.1%. The total RNA was extracted using a RNeasy^®^ Mini Kit and automated RNA purification system (QIAcube; QIAGEN, Hilden, Germany), which includes the procedure of RNase inhibition and DNase treatment. 

### 4.5. Culture of the RHEK Models for the Evaluation of Hydration Factors

Each cup of the RHEK models was placed onto a 24-well culture plate and assay medium was added under the cup. After incubating the plate at 37 °C, 5% CO_2_ atmosphere for 1 day, the RHEK models were treated with a solution of TSE or isolated compounds (final DMSO concentration: 0.1%). Depending on each experiment, the culture time was selected. Namely, RHEK models were cultured for 7 days for TEWL measurement, for 4 days for real-time RT-PCR, and for 5 days for ceramide analysis. The medium containing the test samples was replaced every day.

### 4.6. Measurement of TEWL in RHEK Models

TEWL measurement was performed before treatment and at 1, 3, 5, and 7 days after treatment by Tewitro TW24 (Courage+Khazaka, Cologne, Germany), which can simultaneously measure TEWL in 24-well RHEK models [36,37]. The RHEK model was placed on a thermal insulation mat (HIENAI Mat; Cosmo Bio Co., Ltd., Tokyo, Japan) which keeps the entire bottom surface at 32 °C without a lid for 5 min before measurement. TEWL was measured for 30 min maintaining the bottom surface at 32 °C, and the mean value for the last 10 min was used for the analysis. The measurements were performed under sterile conditions.

### 4.7. Real-Time RT-PCR

The mRNA expression of enzymes related to ceramide synthesis in HaCaT cells and RHEK models were measured by quantitative real-time RT-PCR. After the extraction, 0.1 μg of total RNA was reverse transcribed using PrimeScript™ Reverse Transcriptase to obtain cDNA. Real-time RT-PCR reaction was conducted using TB Green^®^ Premix Dimer Eraser™ and Thermal Cycler Dice^®^ Real-Time System Single (TM 800, Takara Bio Inc.). The specific primers were used as follows: SPT2, 5′-AGCCGCCAAAGTCCTTGAG-3′ as forward and 5′-CTTGTCCAGGTTTCCAATTTCC-3′ as reverse; CerS3, 5′-CCAGGCTGAAGAAATTCCAG-3′ as forward and 5′-AACGCAATTCCAGCAACAGT-3′ as reverse; GCS, 5′-ATGTGTCATTGCCTGGCATG-3′ as forward and 5′-CCAGGCGACTGCATAATCAAG-3′ as reverse; GBA, 5′-TGGCATTGCTGTACATTGG-3′ as forward and 5′-CGTTCTTCTGACTGGCAACC-3′ as reverse; SMS2, 5′-AAGTGTATAACATCAGCTGTGAA-3′ as forward and 5′-CAGTACCAGTTGTGCTAGACTAC-3′ as reverse; ASM, 5′-TGGCTCTATGAAGCGATGG-3′ as forward and 5′-AGGCCGATGTAGGTAGTTGC-3′ as reverse; β-actin, 5′-CATGTACGTTGCTATCCAGGC-3′ as forward and 5′-CTCCTTAATGTCACGCACGAT-3′ as reverse; GAPDH, 5′-AAGGTGAAGGTCGGAGTCAAC-3′ as forward and 5′-GGGGTCATTGATGGCAACAATA-3′ as reverse. The mRNA expression level of each enzyme was determined by the 2-ΔΔCt method and corrected by the expression level of β-actin or GAPDH.

### 4.8. Lipid Extraction and Ceramide Determination

The whole tissue of the RHEK models was carefully peeled from the membrane. The separation of SC was initially carried out by incubation (37 °C, 5% CO_2_ atmosphere) of the tissue in trypsin (2.5 mg/mL)/EDTA (0.25 mg/mL) aqueous solution (1 mL) for 15 min. Then, 10% FBS diluted in PBS (1 mL) was added to stop the trypsin activity and SC separation was carried out under a microscope. The separated SC samples were washed with PBS and stored at −80 °C until the determination of ceramides. The extraction method described in previous studies [32,44] was used for lipid extraction. Namely, the SC samples were homogenized with an ultrasonic homogenizer (AGC Techno Glass Co., Ltd. Shizuoka, Japan) in a 4 mL mixture of chloroform, methanol, and PBS (1:2:0.8). The mixture was centrifuged (3000 rpm, 15 min) at room temperature and the supernatants were collected in test tubes. Chloroform (1 mL) and PBS (1 mL) were added to each supernatant and mixed using a shaker for 20 min. After mixing, the mixture was centrifuged (3000 rpm, 15 min) and the bottom layer was collected using a 1 mL glass syringe with a 22G needle. The collected bottom layer was dried at 30 °C by N_2_ gas blowing. The remaining precipitates were used for the quantification of total protein contents to correct the ceramide contents. The precipitates obtained after lipid extraction were dissolved in a mixture (300 µL) of 10% sodium lauryl sulfate (SDS) and 1 N NaOH (1:9) at 60 °C for 2 h. Then, the mixture was neutralized with 2N HCl (30 µL) and the total protein amounts were determined by the bicinchoninic acid (BCA) method.

The ceramide contents in SC were measured by HPTLC. The quantification method as described by previous studies [32,44] was carried out for TLC analysis. The dried lipid samples were dissolved in a mixture of chloroform and methanol (2:1) and were developed on a TLC plate (10 × 10 cm). The lipid samples were developed twice. Namely, a mixture of chloroform, methanol, and acetic acid (190:9:1) was used for the first development and a mixture of chloroform, methanol, and acetic acid (197:2:1) was used for the second development. After the development, the spots were visualized by 10% copper sulfate in 8% phosphoric acid aqueous solution, followed by heating at 180 °C for 7 min. The spots for the ceramides were scanned and analyzed using an imaging system (ImageQuant LAS500; GE Health Care, CT, USA). The spot areas of ceramides were corrected by the spot areas of ceramide standard.

### 4.9. Compound 48/80- and Histamine-Induced Mice Pruritus (In Vivo Anti-Inflammation Models) 

Male ddY mice aged 5 weeks old were fasted for 18–20 h prior to the experiment. Gum Arabic (5%) in water was administrated orally to the control group. TSE (500 mg/kg), lycoperoside H (10 mg/kg), and DP (30 mg/kg) suspended with 5% gum Arabic in water were orally administered to each group. One hour after administration of the test samples, 3% compound 48/80 solution dissolved in saline was subcutaneously administered (50 µL) to the back of the neck, and the number of scratching behaviors were counted for 30 min [61].

Male ddY mice aged 5 weeks old were fasted for 18–20 h prior to the experiment. The test samples were orally administered to each group. One hour after administration of the test samples, histamine dihydrochloride solution (2 µM) dissolved in saline (50 µL) was subcutaneously administered to the back of the neck, and the number of scratching behaviors were counted for 30 min [62].

### 4.10. PCA Reaction in Mice (In Vivo Anti-Allergy Model) 

Anti-DNP IgE diluted with saline (×2000) was intradermaly administered (20 µL) to both the right and left auricles in male ddY mice aged 8 weeks old. One day after the sensitization, the mice were fasted for 18–20 h. Gum Arabic (5%) in water was administrated orally to the control group. TSE (500 mg/kg) and lycoperoside H (10 mg/kg) suspended with 5% gum Arabic in water were orally administered to each group. Two hours after administration of the test samples, saline containing DNP-BSA (1 mg/mL) and Evans blue (5 mg/mL) was injected (10 mL/kg) into the tail vein. The mice were sacrificed 30 min later, and the auricles were excised. The removed auricles were suspended into 1N KOH (1 mL) and incubated with slow shaking (37 °C, overnight). After incubation, 2 mL of the mixture of 2.5N phosphoric acid and acetone (3:17) was added and centrifuged (2000 rpm, 10 min) to obtain a clean supernatant. The absorbance (620 nm) of the supernatants (200 µL) was measured using a microplate reader (Sunrise RAINBOW, Tecan Group Ltd., Manne Dorf, Switzerland) [63].

### 4.11. Acetic Acid-Induced Writhing and Vascular Permeability in Mice (In Vivo Anti-Inflammation Model) 

Male ddY mice aged 5 weeks old were fasted for 15 h. Gum Arabic (5%) in water was administrated orally to the control group. TSE (500 mg/kg), lycoperoside H (10 mg/kg), tigogenin 3-*O*-β-soratrioside (10 mg/kg), or DX (10 mg/kg) suspended with 5% gum Arabic in water were orally administered to each group. Two percent pontamine sky blue dissolved in saline was injected (10 mL/kg) into the tail vein 55 min after the test samples were administered orally. Five minutes later, 1% acetic acid in saline was injected (10 mL/kg) intraperitoneally and the number of writhings were counted for 15 min. The mice were sacrificed, and the abdomen was immediately opened. After washing of the peritoneal cavity with approximately 8 mL of saline, the washed solution was collected in a glass tube and 0.1 mL of 1N NaOH was added. The solution was topped-up to 10 mL with saline, and the absorbance (590 nm) was measured using a microplate reader to assess the vascular permeability [50].

### 4.12. Glucocorticoid Receptor Competitive Assay (In Vitro Assay for Glucocorticoid Receptor Ligand)

A PolarScreen™ Glucocorticoid Receptor Competitor Assay Kit, Red, was used according to the manufacturer’s protocols. Briefly, lycoperoside H diluted with GR screening buffer to adjust the final concentrations to 0.3, 1, 3, 10, and 30 µM was added to a 96-well black plate. Similarly, DX diluted to adjust the final concentrations to 3, 10, 30, 100, and 300 nM was added. Then, 25 µL of 4× Fluormone™ GS Red solution was added and mixed by shaking on a plate shaker. After shaking, 4× glucocorticoid receptor was added and shaken on a plate shaker. The mixed solutions were incubated in the dark at room temperature for 2 h, and then the fluorescence polarization values were measured at 535 nm (excitation) and 590 nm (emission) using a fluorescence microplate reader (Infinit^®^ 200 PRO, Tecan Group Ltd.).

### 4.13. Histamine-Induced Guinea Pig Tracheal Muscle Contraction (In Vitro Anti-Histaminic Model) 

Male Hartley guinea pigs were anesthetized and sacrificed. The tracheal muscle was harvested and cut horizontally into 5 or 6 strips. One muscle strip was opened and fixed to a Magnus apparatus and placed in Krebs-Henselite solution (NaCl: 117 mM, KCl: 4.7 mM, CaCl_2_: 2.5 mM, MgSO_4_: 1.2 mM, KH_2_PO_4_: 1.2 mM, NaHCO_3_: 24.8 mM, glucose: 11 mM) gently bubbled with 5% CO_2_ and 95% O_2_. 

The organ bath was kept at 37 °C and the specimen was set under a tension of 1 g. Changes in contraction were assessed with an isotonic transducer (Powerlab, AD Instruments, Dunedin, New Zealand) and contractions were recorded (Lab chart 6, AD Instruments). After the tension became stable (30 min), histamine dihydrochloride (final concentration 10 µM) was added to the organ bath to confirm muscle contraction. After washing 5 times with Krebs-Henselite solution, the specimen was allowed to stabilize again. Then, the TSE (final concentration 100 μg/mL) and lycoperoside H (**3**, final concentration 10 μg/mL) diluted in DMSO were added to the bath and the response was assessed for 10 min. Histamine dihydrochloride diluted in buffer (final concentration: 0.1–10 µM) was added cumulatively and the responses were recorded. DP was used as a positive control [63].

### 4.14. Statistics

All the results are expressed as means ± standard error (SE). The significance of the differences was examined by one-way analysis of variance (ANOVA) followed by Dunnett’s test. Bartlett’s test was performed before Dunnett’s test to confirm that all data were parametric. Values of *p* < 0.05 or *p* < 0.01 were considered significant.

## Figures and Tables

**Figure 1 molecules-26-05860-f001:**
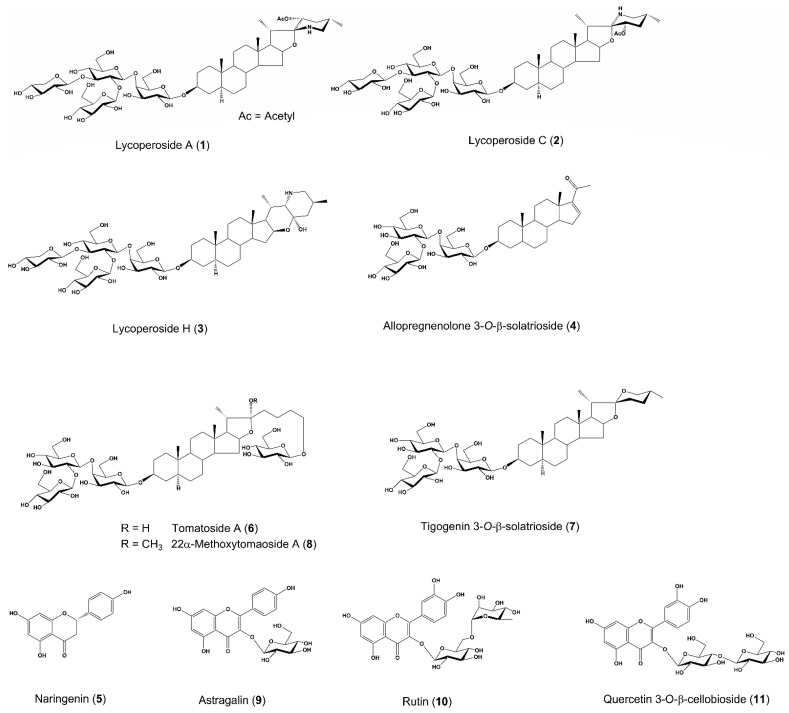
Chemical structures of the isolated compounds (**1**–**11**) from tomato seeds.

**Figure 2 molecules-26-05860-f002:**
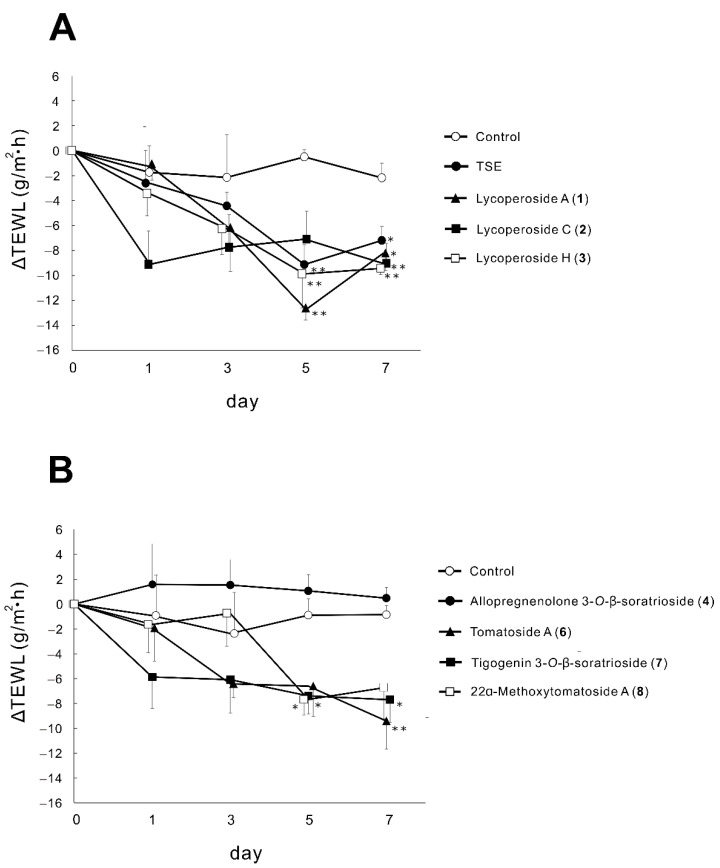
Effects of TSE and tomato seed saponins (**1**–**4**,**6**–**8**) on TEWL in RHEK models. RHEK models were treated with each sample ((**A**) TSE: 10 µg/mL, **1**–**3**: 10 µM, (**B**) **4**–**8**: 10 µM) for 7 days. TEWL measurements were performed on days 0, 1, 3, 5, and 7 using Tewitro TW24. Data are expressed as mean ± SE (*n* = 4). * *p* < 0.05, ** *p* < 0.01 vs. control.

**Figure 3 molecules-26-05860-f003:**
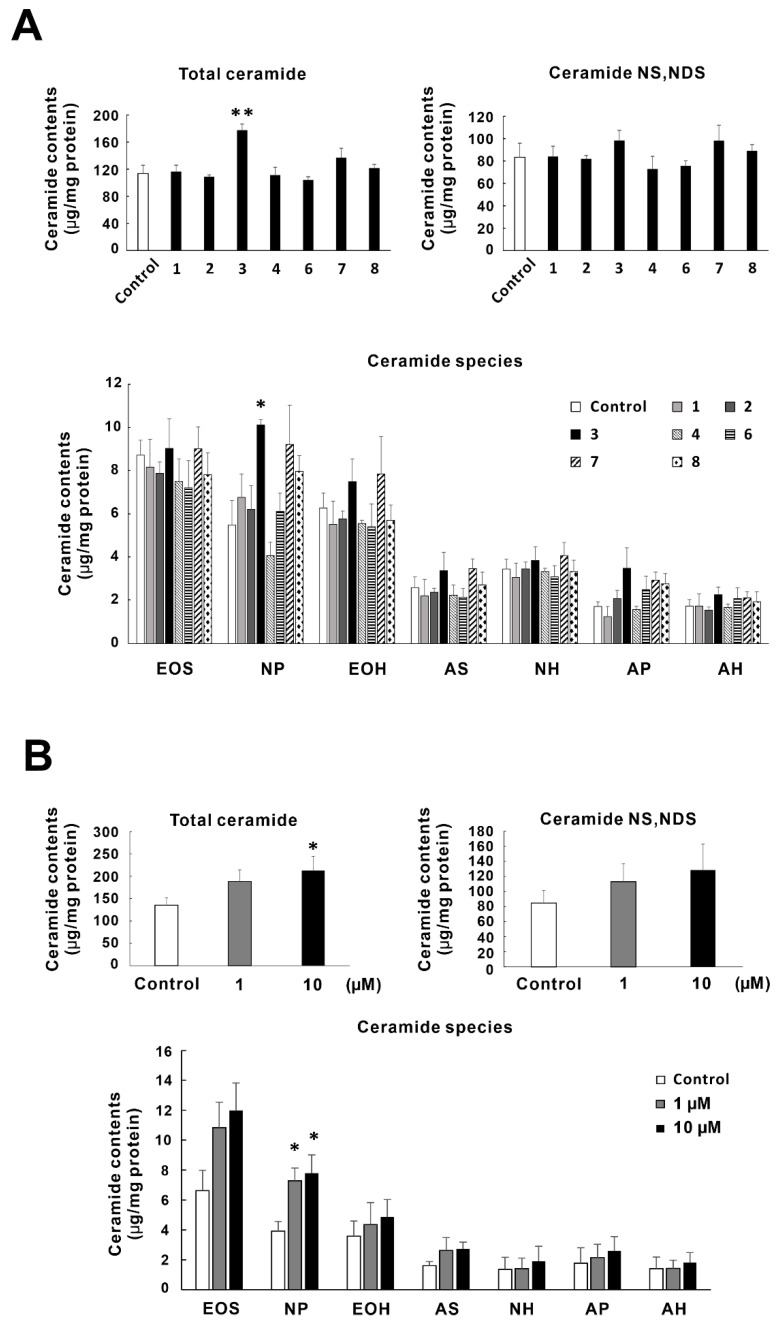
Effects of TSE and tomato seed saponins (**1**–**4**,**6**–**8**) on TEWL in RHEK models. RHEK models were treated for 5 days in culture with 10 µM of tomato seed saponins (**A**) or 1 and 10 µM of lycoperoside H (**B**). The extraction of lipids from the SC of the RHEK model and high-performance thin-layer chromatography (HPTLC) analysis were performed as described in the Materials and Methods section. Data are expressed as mean ± SE (*n* = 4). * *p* < 0.05, ** *p* < 0.01 vs. control.

**Figure 4 molecules-26-05860-f004:**
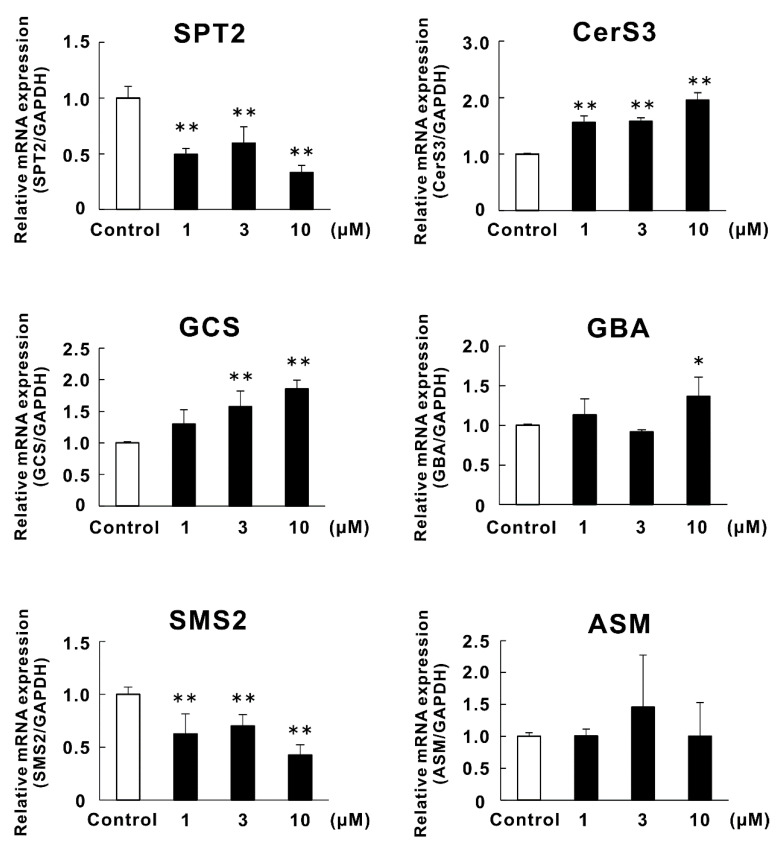
Effects of lycoperoside H (**3**) on the mRNA expression of enzymes related to SC ceramide synthesis. RHEK models were treated for 4 days in culture with 1, 3, and 10 µM of lycoperoside H (**3**). Extraction of total RNA and real-time RT-PCR analysis was performed as described in the Materials and Methods section. Data are expressed as mean ± SE (*n* = 3). * *p* < 0.05, ** *p* < 0.01 vs. control.

**Figure 5 molecules-26-05860-f005:**
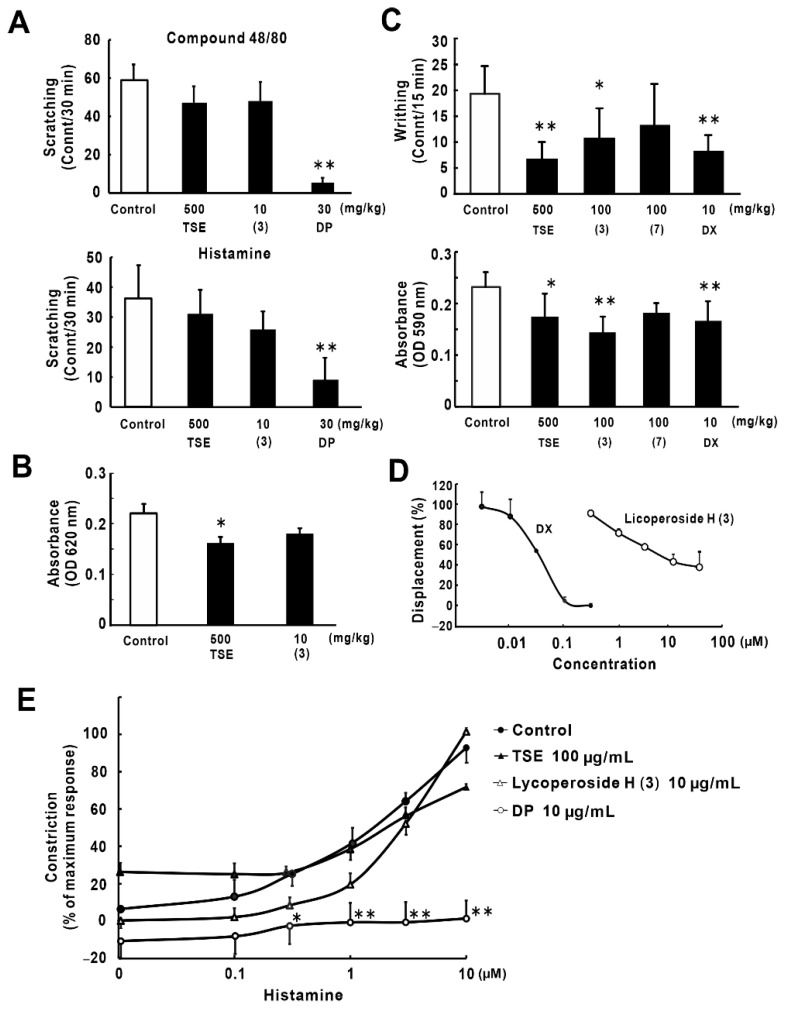
Anti-inflammatory and anti-allergic effects of lycoperoside H (**3**) and its relations to glucocorticoid and histamine. (**A**) Effect on compound 48/80- and histamine-induced scratching behaviors in mice (*n* = 5–7). Pruritus was induced by the subcutaneous administration of compound 48/80 or histamine dihydrochloride. Diphenhydramide hydrochloride (DP) was used as a positive control. (**B**) Effect on the PCA reaction in mice (*n* = 10–14). Both the right and left mouse auricles were sensitized by anti-2,4-dinitrophenyl (DNP). After test sample administration, intravenous administration of 2,4-dinitrophenylated bovine serum albumin (DNP-BSA) and Evans blue was performed, and the pigment that leaked into the auricles was measured. (**C**) Effect on acetic acid-induced writhing behavior and vascular permeability in mice (*n* = 5–7). Two percent pontamine sky blue was intravenously administered to mice after test sample administration. One percent acetic acid was injected intraperitoneally and writhing behavior and leaked pigment into the peritoneal cavity were measured. Dexamethasone (DX) was used as a positive control. (**D**) Glucocorticoid receptor binding ability (*n* = 4). Glucocorticoid receptor competitive assay was performed using a PolarScreen™ Glucocorticoid Receptor Competitor Assay Kit, Red. (**E**) Effect on histamine-induced guinea pig tracheal muscle contraction (*n* = 3–4). An excised tracheal strip was fixed to a Magnus apparatus. Changes in contraction induced by histamine were assessed after test sample addition. DP was used as a positive control. All data are expressed as mean ± SE, * *p* < 0.05, ** *p* < 0.01 vs. control.

**Table 1 molecules-26-05860-t001:** Effects of TSE and isolated compounds (**1**–**11**) on the mRNA expression of proteins related to epidermal hydration in HaCaT cells.

	Relative mRNA Expression				
	Filaggrin	Involucrin	SPT2	CerS3	GCS
Control	1.00 ± 0.03	1.00 ± 0.01	1.00 ± 0.02	1.00 ± 0.02	1.00 ± 0.01
TSE	1.42 ± 0.06	1.56 ± 0.12 **	1.49 ± 0.07 *	1.78 ± 0.04 **	1.62 ± 0.13 **
Lycoperoside A (**1**)	1.47 ± 0.16	0.91 ± 0.04	0.98 ± 0.05	1.26 ± 0.04	1.00 ± 0.04
Lycoperoside C (**2**)	1.91 ± 0.26 **	1.59 ± 0.14 **	1.70 ± 0.21 **	1.72 ± 0.07 **	1.32 ± 0.01 *
Lycoperoside H (**3**)	1.56 ± 0.15	1.67 ± 0.07 **	1.45 ± 0.05	1.50 ± 0.17 **	1.47 ± 0.11 **
Allopregnenolone 3-*O*-β-solatrioside (**4**)	1.57 ± 0.14	1.40 ± 0.06 *	1.01 ± 0.17	1.20 ± 0.06	1.01 ± 0.03
Tomatoside A (**6**)	1.31 ± 0.11	0.96 ± 0.04	1.35 ± 0.09	1.33 ± 0.07 *	1.66 ± 0.15 **
Tigogenin 3-*O*-β-solatrioside (**7**)	1.30 ± 0.13	1.03 ± 0.12	1.59 ± 0.28 *	1.18 ± 0.05	1.58 ± 0.04 **
22α-Methoxytomatoside A (**8**)	1.50 ± 0.09 **	1.21 ± 0.08	1.37 ± 0.09	1.32 ± 0.08 *	1.28 ± 0.08
Naringenin (**5**)	1.30 ± 0.11	1.14 ± 0.12	1.84 ± 0.09 *	1.09 ± 0.08	0.83 ± 0.07
Astragalin (**9**)	1.07 ± 0.09	1.16 ± 0.02	1.82 ± 0.09 **	1.18 ± 0.09	0.81 ± 0.03
Rutin (**10**)	0.80 ± 0.11	1.01 ± 0.13	1.03 ± 0.03	0.66 ± 0.09 **	0.81 ± 0.05 *
Quercetin 3-*O*-β-cellobioside (**11**)	0.72 ± 0.01 *	0.82 ± 0.12	0.79 ± 0.02 *	0.56 ± 0.03 **	0.62 ± 0.03 **

HaCaT cells were treated with TSE (10 µg/mL) or isolated compounds (**1**–**11**, 10 µM) for 6 h. Extraction of total RNA and real-time RT-PCR analysis were performed as described in the Materials and Methods section. Data are expressed as mean ± SE (*n* = 3). * *p* < 0.05, ** *p* < 0.01 vs. control. Each value was corrected by the mRNA expression level of β-actin and shown as the value relative to the control.

## Data Availability

The data that support the findings of this study are available from the corresponding author upon reasonable request.

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
