# Peer review of "Lycoperoside H, a Tomato Seed Saponin, Improves Epidermal Dehydration by Increasing Ceramide in the Stratum Corneum and Steroidal Anti-Inflammatory Effect"

_molecules, 2021, doi:10.3390/molecules26195860_

Round 1
Reviewer 1 Report
they listened to the suggestions, everything Ok
Author Response
Thank you for your review and comments.
Reviewer 2 Report
The authors did not follow any suggestion. In particular, they did not perform new experiments to confirm glucocorticoid-like activity through inhibition of effects of lycoperoside H a glucorticoid receptor antagonist. Also, anti-inflammatory activity of lycoperoside H keeps to be not adequately studied.
Reviewer 3 Report
This paper is a well-written article. This paper deserves publication in molecules.
Author Response
Thank you for your review and comments.
Reviewer 4 Report
- The abstract might be modified with the numerical findings of the study.
- The clear hypothesis of the study will be included at the end of the introduction section.
- The authors are requested to include the ethical details. Why was the experiment executed with two different animal’s sources?
- How about the metabolic pathway of isolated compounds in normal cells? Does the author screen the toxicity profile of the isolated compounds?
- In what basis does the author fix or choose the concentration of TSE (10 μg/mL) or isolated compounds (10 μM) and animal studies (TSE (500 mg/kg), lycoperoside H (10 mg/kg) and diphenhydramine hydrochloride (DP, 30 mg/kg).
- In Histamine-induced guinea pig tracheal muscle contraction studies, the authors mentioned that test substance diluted in DMSO. I am not aware of the test solution. It will be clarified.
- Double-check the abbreviation.
Author Response
Please see the attachment.

This manuscript is a resubmission of an earlier submission. The following is a list of the peer review reports and author responses from that submission.
Round 1
Reviewer 1 Report
In this work, the effects of 11 compounds (including saponins and flavonol glycosides) isolated from tomato seeds were studied on epidermal hydration in immortalized human keratinocytes using a reconstructed human epidermal keratinization models. According to the authors, results obtained with experiments should indicate that among the compounds investigated, lycoperoside H can improve epidermal dehydration and suppresses atopic dermatitis-like inflammation by increasing ceramide in the stratus corneus and through steroid-like anti-inflammatory activity.
The work is interesting, however, some aspects do not make it acceptable in the present form and do not fully support author’s conclusions.
Conclusions on the glucocorticoid activity of lycoperoside H are hasty. To confirm glucocorticoid-like activity it should be shown that the effects of lycoperoside H are inhibited by a glucorticoid receptor antagonist. This experiment should be conducted and serves to clarify the mechanism of action of this compound.
Moreover, anti-inflammatory activity of lycoperoside H is not adequately studied. Acetic acid-induced writhing behavior is more suitable to assess analgesic activity and effects of tlycoperoside H on principal mediators of inflammation should be investigated.
Finally, in the abstract is reported the following sentence “These findings indicate that lycoperoside H can improve epidermal dehydration and suppresses atopic dermatitis-like inflammation by increasing SC ceramide and steroidal anti-inflammatory activity.” This conclusion, related to atopic dermatitis, finds little direct connection with the experiments and turns out to be arbitrary.
Reviewer 2 Report
In general, the subject is not very new, although they have several essays but lack rigor, which translates into a misinterpretation of the results.
For example:
The introduction should be done again, there is no correlation between the topics of the title such as Lycoperoside H AND Epidermal Dehydration AND Ceramide AND Stratum Corneum AND Steroidal Anti-inflammatory Effect
line 362: how did you get the number of animals per group?
line 378: did they use penicillin? fungicide?
Line 392: Is it necessary to explain the extraction of RNA, quality of the RNA?
Line 403: some primers do not have a target according to the first blast
https://www.ncbi.nlm.nih.gov/tools/primer-blast/primertool.cgi?ctg_time=1628194185&job_key=Iyn8ifCF_S3aF20SYHJJIBppWBI3ekMPNg
Other primers have a very large product for qPCR
CBA: 410 bp
https://www.ncbi.nlm.nih.gov/tools/primer-blast/primertool.cgi?ctg_time=1628193783&job_key=AgjdqNGF3C37F0wSQXJoIDtpeRIWemIPFw
line 522: Did you do a normality test? different experiments? triplicate?
Figure 2: Because they normalized by GAPDH and did not perform 2 exp delta delta Cq
Missing references in methods